# Bullying in Elementary Schools: Differences across Countries in the Persian Gulf

**DOI:** 10.3390/children10071108

**Published:** 2023-06-25

**Authors:** Georgios Sideridis, Maisaa Alahmadi

**Affiliations:** 1Boston Childrens’ Hospital, ICCTR, Harvard Medical School, Boston, MA 02115, USA; 2Department of Primary Education, National and Kapodistrian University of Athens, Navarinou 10a, 10680 Athens, Greece; 3Education and Training Evaluation Commission (ETEC), The National Center for Assessment (Qiyas), King Khaled St. Nakheel District, Riyadh 12395, Saudi Arabia; m.ahmadi@etec.gov.sa

**Keywords:** bullying, measurement invariance, alignment, GCC countries

## Abstract

The current research aimed to examine the similarities and differences in bullying prevalence across the six member states of the Gulf Cooperation Council (GCC): Saudi Arabia, Bahrain, Kuwait, Oman, Qatar, and the United Arab Emirates. Level tests require measurement invariance to be met before they can be performed. In 2019, 45k people participated and provided data for the Trends in International Mathematics and Science Study (TIMMS). When the exact measurement invariance (MI) protocols failed, the alignment methodology was used to analyze the data. After ensuring measurement invariance via the free alignment method, findings revealed statistically significant differences in bullying prevalence; specifically, bullying levels were significantly lower in Saudi Arabia compared to all other countries. The United Arab Emirates ranked second, with the second-lowest bullying rates, which were also significantly lower compared to the rates in all the other countries. As a whole, Saudi Arabia had the lowest levels, followed by the UAE, Kuwait, Oman, Bahrain, and Qatar. Although the absolute difference between Saudi Arabia and the other countries was only modest, further research into the causes, consequences, and preventative measures of bullying is warranted.

## 1. Introduction

### 1.1. Bullying in Elementary Schools: Differences across Countries in the Persian Gulf

Positive interactions with one’s peers are crucial to one’s psychological and social growth [1]. Rejection and acceptance by peers have profound effects on a person’s sense of belonging and self-worth. Statistics show that bullying and victimization occur at alarmingly high rates across all demographics (age, gender, culture, and country) [2,3,4,5].

Research has shown that bullying can have a range of negative effects on individuals, including increased feelings of anxiety, depression, and loneliness, as well as decreased academic performance and lower levels of self-esteem [6,7]. These effects can be long-lasting and can impact a person’s future relationships and well-being. In addition to the individual effects of bullying, there are also broader societal consequences. For example, bullying can contribute to a negative school climate, which can in turn lead to increased absenteeism and lower academic achievement [8]. Furthermore, bullying can contribute to a culture of violence and aggression, which can perpetuate itself over time and affect the well-being of entire communities. Given the serious consequences of bullying, it is important for individuals, schools, and communities to take steps to prevent and address bullying behavior. This can involve promoting positive relationships among peers, teaching social and emotional skills, and creating policies and procedures that explicitly prohibit bullying and provide support for those who are affected by it. By taking a proactive approach to bullying prevention, we can create a safer and more inclusive society for all.

Based on international statistics, one out of five students (20.2%) are bullied, and these incidences are more prevalent in males with regard to physical means, but in females more so with regard to exclusion and indirect bullying behaviors [9]. Of particular concern are bullying behaviors delivered via electronic means, using social media and easily accessible multitask phones. Recent studies have noticed that these cyberbullying behaviors, although they were primarily prevalent in adolescence, are now increasingly observed in primary schools [10]; thus, the age of prevalence has significantly lowered, raising concerns over spillover effects in the elementary grades.

In particular, the increasing prevalence of cyberbullying in primary schools is a worrying trend that highlights the need for early intervention and prevention efforts. Cyberbullying can take many forms, such as spreading rumors online, sharing embarrassing photos or videos, and sending threatening or insulting messages. Unlike traditional forms of bullying, cyberbullying can occur at any time and in any location, making it difficult for children and parents to escape from its effects. Research has shown that cyberbullying can have serious negative effects on young children’s emotional and psychological well-being, including increased levels of anxiety, depression, and social isolation [11]. Moreover, cyberbullying can lead to long-term consequences, such as lower academic achievement and reduced opportunities for social interaction and personal development [12].

### 1.2. Importance and Goals of the Present Study

Risky behaviors [13], physical aggression [14], sexual abuse [15], negative educational outcomes [16], moral reasoning [17], and even psychopathological tendencies have all been linked to bullying, making it clear that this aggressive behavior has far-reaching consequences for healthy emotional development [18]. As a result, valid and reliable measurement is necessary for assessing and intervening on a social phenomenon with far-reaching effects on children’s well-being, and was the first goal of the present study as a means of validating the existing bullying scale that is utilized in the TIMSS for the measurement of aberrant student behaviors. The current study aimed to examine the similarities and differences in bullying prevalence across the six member states of the Gulf Cooperation Council (GCC): Saudi Arabia, Bahrain, Kuwait, Oman, Qatar, and the United Arab Emirates. The selection of the GCC countries was based on the following reasons: (a) the GCC is grounded in the political and economic reasoning of Middle Eastern countries that share similar values, culture, language, and origin. All these countries share the importance of family, community engagement, respect for others and for social norms, strong religious values and adherence to the Islamic law (Sharia). All these countries also share similar academic achievements based on the TIMSS by being ranked lower than average in mathematics and science. However, these similarities do not preclude differences that may be a function of economic growth, societal changes, and new developments in each country. In particular, all countries value education and invest heavily in it, but the qualitative aspects of student life, such as student behaviors and norms at school are not necessarily similar. For example, [19] reported that 26% of Saudi students reported being bullied, but relevant data are missing from other Gulf countries. To this end, investigating levels of bullying may reflect a deviation from societal values and norms, respect, and kindness. Consequently, this second objective will be reached after a comprehensive psychometric analysis of the bullying scale using the TIMMS 2019 cohort for fourth-grade students.

## 2. Methods

### 2.1. Participants and Procedures

Participants were 44,977 4th-grade students who participated in the 2019 TIMMS assessment. To facilitate comparisons between the Kingdom of Saudi Arabia and the other Gulf countries’ primary education systems, we zeroed in on data from the Gulf countries and the 4th grade cohort. From Bahrain, 4929 students took part; from Kuwait, 3463; from Oman, 5731; from Qatar, 4113; from the UAE, 22,867; and from Saudi Arabia, 3874. There were about as many males as females (49.9% females and 50.1% males) in the sample as a whole. Sample sizes were large enough to permit using only participants with complete data. Sample selection per country involved a two-stage random sampling approach, with eligible schools selected first and then intact classes being selected from the sampled schools again using random assignment (for specific information on sampling, the reader is directed to the TIMSS technical manual at: https://nces.ed.gov/timss/timss15technotes_intlreqs.asp) (accessed on 22 June 2023). Given variability on how grades are defined per country, it was ensured that students were at least 9.5 years old at the time of testing. Student exclusionary criteria involved the presence of an intellectual or functional disability and students’ inability to read or speak the language of the test. The administration of the measures was conducted by National Testing Centers under the supervision of the TIMSS research team, the International Association for the Evaluation of Educational Achievement (IEA; https://nces.ed.gov/programs/coe/indicator/cnt/intl-grades-4-8-math-science) (accessed on 22 June 2023), and Pearson publishing (https://www.pearson.com/uk/web/timss/about-timss.html) (accessed on 22 June 2023) to ensure the provision of test materials and the application of a standardized protocol of assessment. The studies in each country were approved by the TIMSS Ethical Review Board, as well as the relevant bodies in each country. Informed consent, anonymity and confidentiality, voluntary participation, and data protection procedures are described thoroughly on the TIMSS website. 

### 2.2. Measures

The measure comprised an 11-item Likert type instrument that evaluated bullying behaviors. The items were as follows: (1) Made fun of me or called me names; (2) Left me out of their games or activities; (3) Spread lies about me; (4) Stole something from me; (5) Damaged something of mine on purpose; (6) Hit or hurt me; (7) Made me do things I didn’t want to do; (8) Sent me nasty or hurtful messages online; (9) Shared nasty or hurtful things about me online; (10) Shared embarrassing photos of me online; and (11) Threatened me. The scaling system involved a frequentist 4-point scaling as follows: Never; A few times a year; Once or twice a month; and At least once a week. The specific instrument is considered current and of proper content validity, as it includes behaviors specific to cyberbullying and electronic means. The TIMSS data analytic group considered the measure to be unidimensional and provided a total score for its measurement for the 4^th^-grade elementary school group. We tested this proposition using Exploratory Factor analytic procedures, as well as the bifactor model, to ensure the optimal factor structure of the bullying instrument prior to proceeding with tests of measurement invariance.

### 2.3. Data Analyses

#### Prerequisite Analyses: Tests of Measurement Invariance

It is crucial to verify the validity of measurement invariance (MI) before proceeding with level tests. Simply put, this presumption implies that the structures being measured are equivalent across groups, so that observed level differences reflect genuine distinctions rather than the effects of measurement inconsistencies. The first stage of this procedure is to verify a simple structure that holds across all groups, a process known as configural invariance. This process included estimating omega reliability and fitting a unidimensional model to the data. Strict measurement invariance was then applied, where estimates of simple structure, factor loadings, and item intercepts were standardized to be the same in all six countries. Despite the availability of other methodologies, such as approximate measurement invariance using Bayesian estimators, we opted to examine measurement invariance using the alignment methodology [20] due to the model’s sensitivity to rejection in the presence of a large sample size such as the current one.

As [21] pointed out, there is a major difference between approximate measurement invariance and the alignment procedure. With approximate MI, the model seeks a solution where the variance of the measurement parameters is small. With the proposed alignment method, however, the goal is to find a solution that includes measurement parameters “with a large degree of minor non-invariance” (p. 7). Because the alignment methods engage the configural model only with no further constraints, the model attempts to maximize invariance through allowing factor means and variances to vary across groups. Models involved modeling the Likert-type items as categorical ordered indicators of each measured behavior. All analyses were conducted using Mplus 8.10, Lavaan, and the Measurement Invariance Explorer (MIE) package in R.

## 3. Results

### 3.1. Testing Simple Factor Structure and Internal Consistency Reliability of the Bullying Scale

Prior to testing measurement invariance, it was important to test and verify the simple structure of the bullying scale of the TIMSS with the fourth-grade population, particularly in light of evidence from the eighth-grade data that cyberbullying comprised a second dimension. We therefore tested the unidimensional model, which provided a good model fit to the data. Specifically, descriptive fit indices were acceptable (CFI = 0.973, TLI = 0.979), and unstandardized residuals also <8% (i.e., RMSEA = 0.058, C.I._95%_ = 0.057–0.058). The omnibus chi-square test statistic was significant, but this was expected given the large sample size of more than 44k participants. We further tested the presence of additional dimensions using an Exploratory Factor Analysis model using Maximum Likelihood Estimation and quartimax rotation using Kaiser’s normalization, as facets of bullying would likely be significantly correlated. This model suggested that a second factor could potentially be plausible with the three cyberbullying items; however, all three items had factor loadings that were substantially lower compared to those loading on the first factor (i.e., 0.494/0.615, 0.593/0.612, and 0.500/0.578), all favoring the one dimension solution. Given the inconclusive results observed from the EFA model, we further proceeded with testing a bifactor model in which bullying, cyberbullying, and global-domain bullying dimensions were operative. Results after fitting the bifactor model indicated that the global bullying domain factor loadings ranged between 0.570 and 0.860, whereas all the factor loadings of the two bullying domains had factor loadings ranging between 0.157 and 0.453, suggesting that the global factor was dominant. Consequently, the single-domain bullying structure was assumed to be the optimal factor structure with these data from the fourth-grade students in the TIMSS.

Furthermore, the internal consistency of the instrument using Omega coefficient was 0.882, which is excellent, confirming the unidimensional structure of the scale.

### 3.2. Tests of Strong Measurement Invariance: Traditional Approach

The next step involved specifying tests of the equality of slope terms across countries (metric invariance) followed by tests of the equality of intercepts (scalar invariance). Upon satisfaction of both prerequisite assumptions, tests of latent means can be conducted. Tests of exact fit showed a failure of the metric model. Specifically, based on the global chi-square test results showed significant misfit from constraining the slopes of the 11 items across all six countries [Difference χ^2^(55) = 240.383, *p* < 0.001]. Furthermore, the difference between metric and scalar models was again significant, pointing to significant misfit in the chi-square statistic from constraining item intercepts across countries [Difference χ^2^(50) = 668.959, *p* < 0.001]. These results rendered the comparison of latent means meaningless, unless some form of measurement invariance, partial, or other approach was achieved. For this purpose, the alignment procedure was utilized as shown below.

### 3.3. Tests of Measurement Invariance: Fixed- and Free-Alignment Methodologies

As described above, the alignment procedure was utilized to test for the equivalence of simple structures. The procedure involves two means: a freely estimated one and one, termed “fixed”, where a reference group is specified to have a mean of zero. The authors of [22] recommend always starting with the fixed alignment method, which will most likely result in decreased standard errors compared to the free method. Table 1 displays the results from applying the fixed method alignment procedure. As shown in Table 1, 42 out of the 198 estimated parameters were non-invariant. This number shows a lack of invariance in 21.2% of the tested invariance parameters. Ref. [22] stated: “A rule of thumb is that as long as the number of non-invariant parameters is less than 20%, we can expect the alignment method to work correctly” (p. 6). The fixed alignment methodology obviously violated this rule of thumb; thus, we proceeded with the free alignment option. Free alignment works best if there is substantial non-invariance. These results are shown in Table 2. As shown in the table, there were 36/198 non-invariant parameters, amounting to 18.9% of the total number of tests. Further evidence was provided by investigating the values of the R-square statistic as a degree of non-invariance. The average number of R-square values across estimated parameters was 0.615, which is quite high, although rules of thumb are not currently available and also given that small R-square values do not necessarily imply non-invariance (such as when the levels and variability of an estimate are relatively low). Mean estimates of group invariance were 5.02, suggesting that on average, five out of the six countries were invariant across all tests of intercepts and slopes. Last, to further conclude that the free alignment procedure was successful, we explored the measurement invariance using the MIE package in R through exploring the presence of clusters of groups.

After transforming the factor loadings and intercept parameters across countries to distances using a multidimensional scaling framework, we used the visual analysis provided by [23]. Then, the distances were considered to be within a 0.01-unit range. Both the presence of subgroups and that of outlying groups are compatible with the method (countries). Distances between countries calculated from CFI (upper panel) and RMSEA (lower panel) values for a comparison of configural and metric models are shown in Figure 1. (lower panel). Distances between countries on RMSEA (upper panel) and CFI (lower panel) based on fitted models when contrasting metric and scalar models are shown in Figure 2. There are minimum distances between countries on each estimate that fall within a hypothetical elliptical shape, indicating invariance. Estimates of the CFI and RMSEA for comparing configural–metric–scalar models across countries are shown in Table 3, Table 4, Table 5 and Table 6. Difference values were consistently less than 0.01, as shown in the right table columns, indicating minimal non-invariance.

### 3.4. Latent Mean Differences across Gulf Countries

After ensuring measurement invariance with the free alignment method, the results indicated statistically significant level differences in bullying rates between Gulf countries (see Table 7). Saudi Arabia, in particular, had the lowest bullying rates of any country surveyed. The United Arab Emirates ranked second lowest, with rates significantly lower than those of any country except Saudi Arabia. Last but not least, we found that bullying incidence rates in Qatar were noticeably higher than those in Oman, Kuwait, the UAE, and Saudi Arabia. Thus, Saudi Arabia topped the list for lowest bullying rates, followed by the UAE, Kuwait, Oman, Bahrain, and finally Qatar. According to effect size recommendations [24], the differences that were found to be significant were relatively small, with the largest difference (between Saudi Arabia and Qatar) reflecting a small-to-medium effect (i.e., between 0.2–0.5).

## 4. Discussion

The purpose of the present study was to compare and contrast levels of bullying in the six Gulf countries, namely Saudi Arabia, Bahrain, Kuwait, Oman, Qatar, and the United Arab Emirates. First, the psychometric properties of the bullying scale were investigated, followed by tests of latent means. Several important findings emerged. In terms of the psychometric analyses of the bullying scale, the results indicated very good model fit as per the unidimensional structure of the eleven bullying behaviors using data from all six Gulf countries. Internal consistency estimates were also acceptable. Thus, the 11-item measure utilized in TIMMS 2019 using the fourth-grade cohort possesses desirable psychometric properties to validly assess bullying behaviors, reflecting a single latent factor.

When compared to the other Gulf states, bullying rates were significantly lower in Saudi Arabia, the study’s focus country. Although there were notable differences in levels, the effect sizes observed were small. Several observations are in order here for the notable differences in bullying favoring the Kingdom of Saudi Arabia compared to the rest of the Gulf countries. First, the culture in Saudi Arabia emphasizes respect and obedience towards authority, as well as interpersonal behaviors and interactions that conform to cultural and religious values [25,26]. The Saudi Arabian culture also places a heavy emphasis on collectivism, family, and community values, reflecting the idea of “wasta”, which refers to the utilization of social networks and personal relationships to acquire an advantage in personal and professional contexts [27]. Because forming relationships with others and maintaining peace within groups reflects a main goal of individuals, bullying may be less prevalent in Saudi Arabia so that individuals will not act in ways that would harm or damage their social relationships [28]. Kindness, respect, empathetic understanding, and compassion toward others are fostered by the cultural and religious values of the Kingdom, which may in turn discourage bullying behaviors [29,30,31]. These values are cultivated through parental involvement in their children’s education as parents target creating a sense of accountability and responsibility in their children so that they behave respectfully and appropriately. Furthermore, there is evidence that the Saudi government implements programs to increase awareness between students, staff, teachers, and parents on both how to identify and also how to address bullying instances [32,33]. Last, a potential explanation lies in the imposition of severe and long-lasting consequences for bullies, which can have deleterious effects for the development and future prospects [33]. For instance, the Saudi Arabian Ministry of Education published a new anti-bullying policy in 2014 that mandates schools take disciplinary action against bullies, such as suspension, expulsion, or even reporting to the authorities [34]. Last but not least, gender segregation may lessen occurrences of bullying between genders, since there may not be any amicable or amorous connections that might cause rivalry and conflict.

There are several caveats to the present investigation as well. To begin, several of the inferential statistical findings exhibit extremely high powers and, despite surpassing conventional levels of significance, may not be substantial in and of themselves. Second, there was a lack of explanatory variables that could illustrate and explain the observed bullying rates. While several academic variables are accessible in the TIMMS data, the same cannot be said for variables pertaining to students’ social, emotional, or physical well-being. As a result, this detail was absent from the available data. Despite these limitations, the current study sheds light on the prevalence of bullying in Gulf countries and highlights the need for ongoing school-based initiatives to prevent and address bullying behavior. Teachers and administrators can develop effective strategies to provide secure and supportive learning environments for all students by identifying factors that contribute to low levels of bullying, such as cultural values, strict disciplinary procedures, and parental involvement.

### 4.1. Implications of the Present Findings for Practice

Given the serious consequences of cyberbullying, it is important for parents, educators, and policy-makers to work together to prevent and address this behavior. This can involve educating children about appropriate online behavior, providing support and resources for those who have been affected by cyberbullying, and establishing clear guidelines and consequences for cyberbullying in schools and other settings. Furthermore, it is important to recognize that cyberbullying is not a problem that can be solved by any one group alone but requires multidisciplinary teams for evaluation and prevention.

### 4.2. Conclusions and Future Directions

It is concluded that levels of bullying were significantly lower in Saudi Arabia compared to all other GCC countries, but were also reflective of small effect sizes.

There are various directions that research can go in the future. Studies that examine predictors of bullying behaviors and their differentiation across age and gender groups will likely inform treatment and interventions. Enriching the current instrument with more behaviors (as in the eighth-grade cohort in the TIMSS) will also address issues of content validity and dimensionality. Last, the impact of other factors, such as mental health support and social–emotional learning, on bullying prevention and student well-being should be studied.

## Figures and Tables

**Figure 1 children-10-01108-f001:**
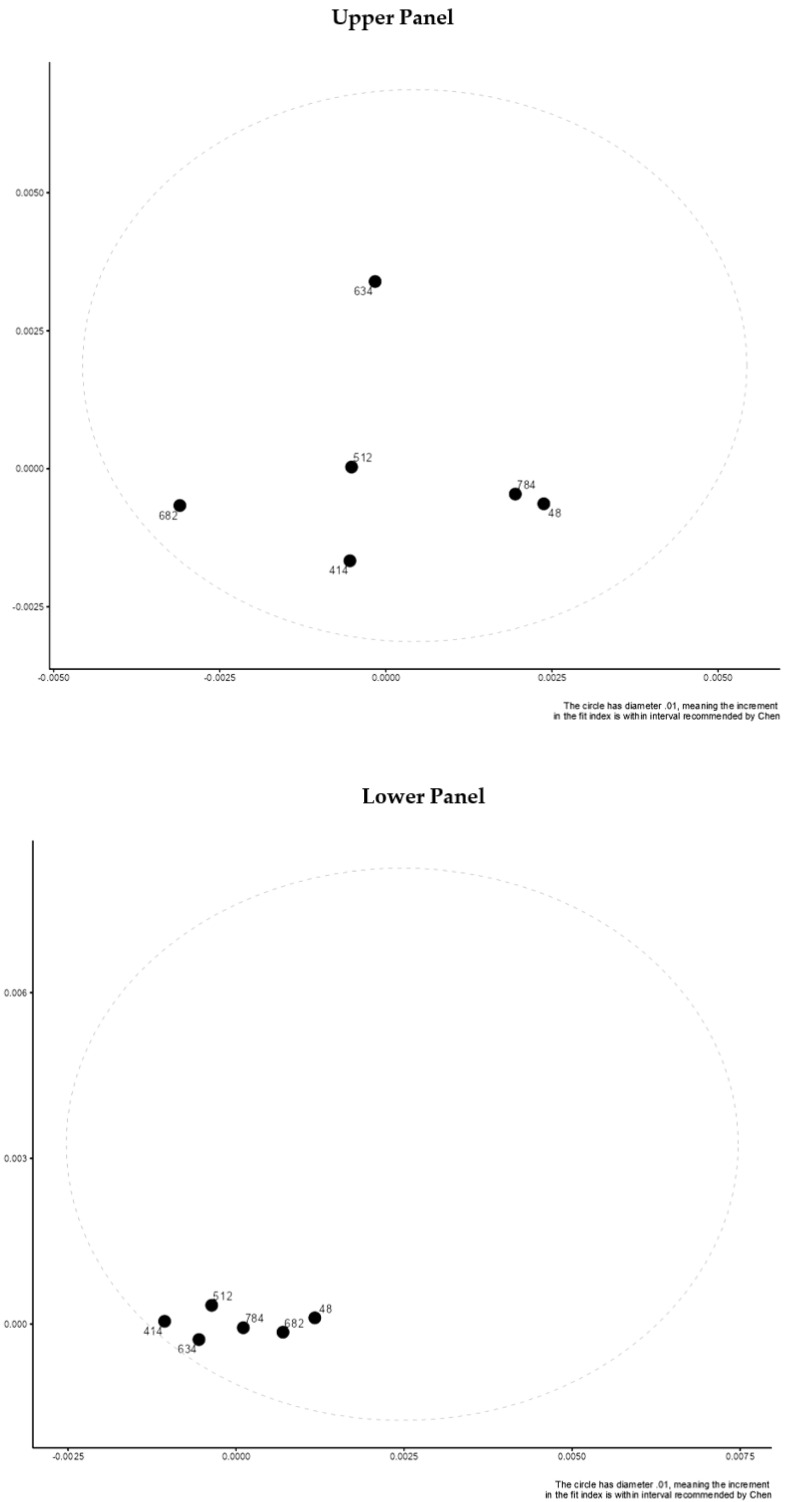
Distances between countries on RMSEA (upper panel) and CFI (lower panel) based on fitted models when contrasting configural and metric models.

**Figure 2 children-10-01108-f002:**
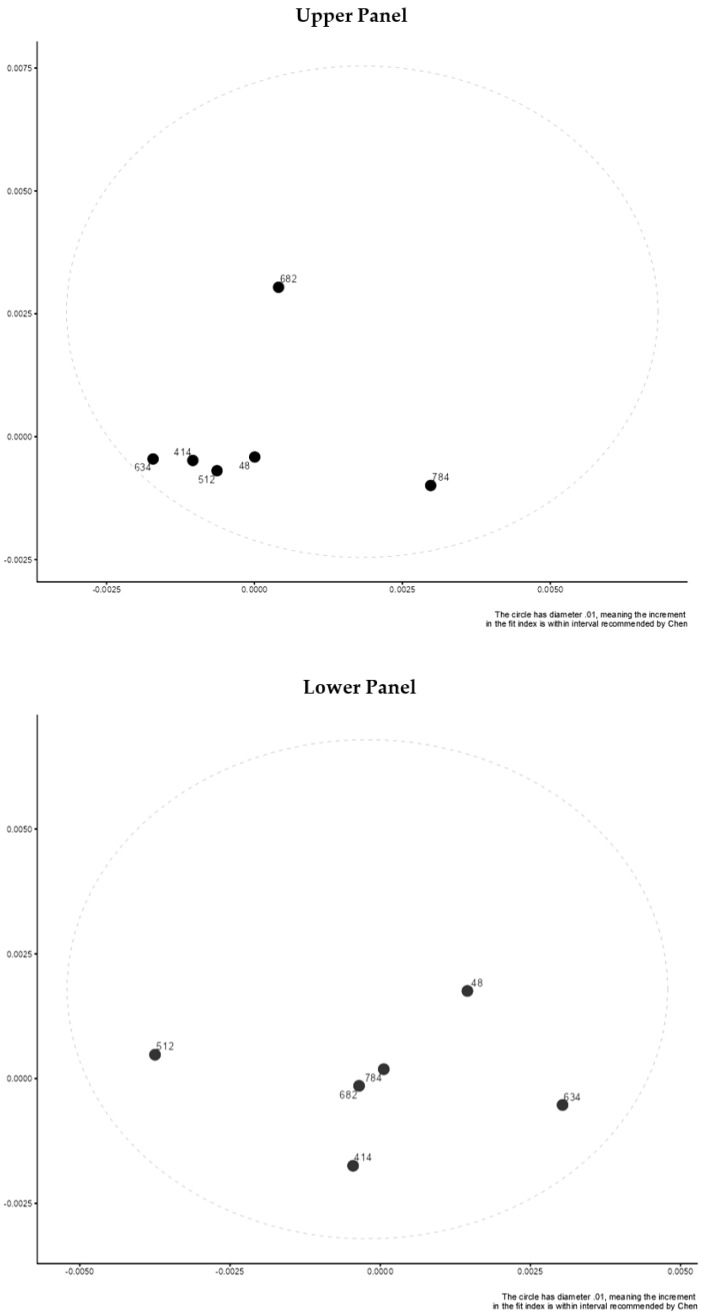
Distances between countries on RMSEA (upper panel) and CFI (lower panel) based on fitted models when contrasting metric and scalar models.

**Table 1 children-10-01108-t001:** Alignment procedure of bullying scale across Gulf countries using Mplus 8.10.

Intercepts/Thresholds of Bullying Items	
V1$1	784 48 (414) (512) 634 (682)
V1$2	784 48 414 (512) 634 682
V1$3	784 48 414 (512) 634 682
V2$1	784 (48) 414 512 634 682
V2$2	(784) (48) 414 512 634 682
V2$3	784 48 414 (512) 634 (682)
V3$1	784 48 414 (512) 634 (682)
V3$2	784 48 414 (512) 634 682
V3$3	784 48 414 512 634 682
V4$1	(784) (48) 414 512 634 682
V4$2	784 48 414 512 634 682
V4$3	784 48 414 (512) 634 682
V5$1	784 48 414 (512) 634 682
V5$2	784 48 (414) (512) 634 682
V5$3	784 48 (414) (512) 634 682
V6$1	784 48 414 (512) 634 682
V6$2	784 48 414 (512) 634 682
V6$3	(784) 48 414 (512) 634 682
V7$1	(784) 48 414 512 (634) 682
V7$2	784 48 (414) 512 (634) 682
V7$3	784 48 414 512 634 682
V8$1	784 48 414 (512) 634 682
V8$2	784 48 414 512 634 682
V8$3	784 48 414 512 634 682
V9$1	(784) 48 414 512 634 682
V9$2	(784) 48 414 512 634 682
V9$3	784 48 414 512 634 682
V10$1	(784) 48 414 512 634 682
V10$2	(784) 48 414 512 634 682
V10$3	784 48 414 512 634 682
V11$1	784 (48) 414 512 634 682
V11$2	784 (48) 414 512 634 682
V11$3	(784) 48 414 512 634 682
Factor Loadings for Unidimensional Bullying Scale	
V1	784 48 414 512 634 682
V2	(784) (48) 414 512 634 682
V3	784 48 414 512 634 682
V4	784 48 414 512 634 682
V5	784 48 (414) (512) 634 682
V6	784 48 414 (512) 634 682
V7	784 48 (414) 512 634 682
V8	784 48 414 512 634 682
V9	(784) 48 414 512 634 682
V10	(784) 48 414 512 634 682
V11	784 48 414 512 634 682

Note: Estimates in parentheses indicate non-invariance across countries. The codes for countries are as follows: 784 = United Arab Emirates; 48 = Bahrain; 414 = Kuwait; 512 = Oman; 634 = Qatar; 682 = Saudi Arabia.

**Table 2 children-10-01108-t002:** Free-alignment procedure of bullying scale across Gulf countries.

Intercepts/Thresholds	
V1$1	784 48 (414) (512) 634 682
V1$2	784 48 414 (512) 634 682
V1$3	784 48 414 (512) 634 682
V2$1	784 48 (414) (512) 634 682
V2$2	784 48 414 512 634 682
V2$3	(784) 48 414 512 634 (682)
V3$1	784 48 (414) (512) 634 (682)
V3$2	784 48 414 (512) 634 682
V3$3	784 48 414 512 634 682
V4$1	(784) (48) 414 512 634 682
V4$2	(784) 48 414 512 634 682
V4$3	784 48 414 512 634 682
V5$1	(784) 48 414 512 634 682
V5$2	784 48 414 512 634 682
V5$3	784 48 414 (512) 634 682
V6$1	(784) 48 414 512 634 682
V6$2	784 48 414 512 634 682
V6$3	(784) 48 414 512 634 682
V7$1	(784) 48 414 512 (634) 682
V7$2	784 48 414 512 634 682
V7$3	784 48 414 512 634 682
V8$1	784 48 414 (512) 634 682
V8$2	784 48 414 512 634 682
V8$3	(784) 48 414 512 634 682
V9$1	784 (48) 414 512 634 682
V9$2	784 48 414 512 634 682
V9$3	784 48 (414) 512 634 682
V10$1	784 48 (414) (512) 634 682
V10$2	784 48 (414) (512) 634 682
V10$3	784 48 (414) (512) 634 682
V11$1	784 48 414 512 634 682
V11$2	(784) 48 414 512 (634) 682
V11$3	(784) 48 414 512 (634) 682
Loadings for F1	
V1	784 48 414 512 634 682
V2	(784) (48) 414 512 634 682
V3	784 48 414 512 634 682
V4	784 48 414 512 634 682
V5	784 48 (414) (512) 634 682
V6	784 48 414 (512) 634 682
V7	784 48 (414) 512 634 682
V8	784 48 414 512 634 682
V9	(784) 48 414 512 634 682
V10	(784) 48 414 512 634 682
V11	784 48 414 512 634 682

Note: Estimates in parentheses indicate non-invariance across countries. The codes for countries are as follows: 784 = United Arab Emirates; 48 = Bahrain; 414 = Kuwait; 512 = Oman; 634 = Qatar; 682 = Saudi Arabia.

**Table 3 children-10-01108-t003:** Differences between countries on estimates of CFI when contrasting configural and metric models.

Group 1	Group 2	Configural	Metric	Difference
784	48	0.885	0.884	0.001
784	414	0.885	0.884	0.001
784	512	0.877	0.876	0.001
784	634	0.88	0.88	0.001
784	682	0.879	0.879	0.000
48	414	0.879	0.877	0.002
48	512	0.855	0.854	0.001
48	634	0.866	0.864	0.002
48	682	0.859	0.859	0.000
414	512	0.887	0.887	0.000
414	634	0.899	0.898	0.000
414	682	0.894	0.892	0.002
512	634	0.872	0.872	0.001
512	682	0.866	0.865	0.001
634	682	0.877	0.876	0.001

Note: The codes for countries are as follows: 784 = United Arab Emirates; 48 = Bahrain; 414 = Kuwait; 512 = Oman; 634 = Qatar; 682 = Saudi Arabia.

**Table 4 children-10-01108-t004:** Differences between countries on estimates of RMSEA when contrasting configural and metric models.

Group 1	Group 2	Configural	Metric	Difference
784	48	0.109	0.104	0.005
784	414	0.109	0.104	0.005
784	512	0.11	0.104	0.006
784	634	0.112	0.106	0.006
784	682	0.112	0.106	0.006
48	414	0.109	0.104	0.005
48	512	0.111	0.105	0.005
48	634	0.117	0.111	0.005
48	682	0.117	0.111	0.006
414	512	0.099	0.093	0.005
414	634	0.104	0.099	0.005
414	682	0.104	0.100	0.005
512	634	0.107	0.102	0.005
512	682	0.107	0.102	0.005
634	682	0.113	0.108	0.005

Note: The codes for countries are as follows: 784 = United Arab Emirates; 48 = Bahrain; 414 = Kuwait; 512 = Oman; 634 = Qatar; 682 = Saudi Arabia.

**Table 5 children-10-01108-t005:** Differences between countries on estimates of CFI when contrasting metric and scalar models.

Group 1	Group 2	Metric	Scalar	Difference
784	48	0.884	0.883	0.001
784	414	0.884	0.883	0.001
784	512	0.876	0.874	0.002
784	634	0.88	0.879	0.000
784	682	0.879	0.878	0.000
48	414	0.877	0.873	0.004
48	512	0.854	0.849	0.005
48	634	0.864	0.862	0.002
48	682	0.859	0.857	0.002
414	512	0.887	0.884	0.004
414	634	0.898	0.895	0.003
414	682	0.892	0.891	0.001
512	634	0.872	0.865	0.007
512	682	0.865	0.863	0.002
634	682	0.876	0.874	0.002

Note: The codes for countries are as follows: 784 = United Arab Emirates; 48 = Bahrain; 414 = Kuwait; 512 = Oman; 634 = Qatar; 682 = Saudi Arabia.

**Table 6 children-10-01108-t006:** Differences between countries on estimates of RMSEA when contrasting metric and scalar models.

Group 1	Group 2	Metric	Scalar	Difference
784	48	0.104	0.099	0.005
784	414	0.104	0.099	0.005
784	512	0.104	0.100	0.004
784	634	0.106	0.101	0.005
784	682	0.106	0.101	0.005
48	414	0.104	0.101	0.003
48	512	0.105	0.102	0.003
48	634	0.111	0.107	0.004
48	682	0.111	0.107	0.004
414	512	0.093	0.090	0.003
414	634	0.099	0.095	0.003
414	682	0.100	0.095	0.004
512	634	0.102	0.099	0.002
512	682	0.102	0.098	0.004
634	682	0.108	0.104	0.004

Note: The codes for countries are as follows: 784 = United Arab Emirates; 48 = Bahrain; 414 = Kuwait; 512 = Oman; 634 = Qatar; 682 = Saudi Arabia.

**Table 7 children-10-01108-t007:** Latent mean differences across Gulf countries in bullying based on TIMMS data.

Country	Country Code	Factor Mean	Groups with Significantly Smaller Factor Mean			
Qatar	634	0.241	414	512	784	682
Bahrain	48	0.223	512	784	682	
Kuwait	414	0.171	784	682		
Oman	512	0.168	784	682		
United Arab Emirates	784	0.087	682			
Saudi Arabia	682	0.000				

Note: The codes for countries are as follows: 784 = United Arab Emirates; 48 = Bahrain; 414 = Kuwait; 512 = Oman; 634 = Qatar; 682 = Saudi Arabia.

## Data Availability

Data are available from the official study of TIMSS 2019.

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
