# Peer review of "Bullying in Elementary Schools: Differences across Countries in the Persian Gulf"

_children, 2023, doi:10.3390/children10071108_

Round 1
Reviewer 1 Report
I would like to say thanks for the opportunity to review this article.
The article presented has a very important and actual theme with relevance for the improvement of health interventions.
It presents the perspective of bullying prevalence in GCC Countries, allowing to know what could be important to invest in, in terms of interventions programmes.
Overall, the article has a scientific and appropriate writing, including all the components of a good scientific research. The title and abstract are related with the content. The keywords are linked to the research, but only one is indexed which can turn the article more hidden in data bases.
In terms of writing, there are some acronyms and abbreviations that don’s have the full text in their first use (i.e. TIMMS, MI, MIE, ...). there are some small flaws in the writing, and for that text should be revised carefully (i.e. “heir” in line 253).
Introduction allows the framing of the theme and the research itself, but is poor and lacks some understanding of the theme. The main goal is appropriate. Methodology is scientifically appropriate, despite some lack of information. There is not clear some methodology options and procedures regarding the sample selection and data collection. It is not clear which data methodology was used to select the sample and what were the inclusion and exclusion criteria to select the sample. It is not clear also what methodology was used to collect data, how was the questionnaire applied, what questions besides the instrument to evaluate bullying were integrated in the questionnaire and how does the instrument to evaluate bullying was scored.
Results are adequate, and it was appropriate to include some information about sociodemographic data, and about results of the bullying questions. It is not possible to know the media or any information about any of the questions or bullying in general in any country, because it is only presented the mean differences. Being the purpose to understand the phenomena, it is important to present more specific results, like means and frequencies. Table 6 need to be shaped correctly. Discussion is done according to the results of the study, allowing a comparison and analysis with the scientific current evidence. Authors present limitations and strengths but conclusions are missing.
References are pertinent, adequate, but being such an important and recent theme, it was important to include more recent references (more than 50% have more than 5 years and almost 30% have more than 10 years). Citation style used is not according to the requested.
For this, we suggest the following corrections:
- Text should be revised
- Introduction should be improved
- Keywords should be revised, being reduced and indexed always as possible
- Methodology should be completed, and completely described
- Results should be completed.
- A conclusion should be included.
- Citation style need to be corrected, and references revised (there are some flaws)
Thank you.
there are some small flaws in the writing, and for that text should be revised carefully (i.e. “heir” in line 253).
Reviewer 2 Report
Dear Author,
I congratulate your effort to collect the data from a large sample and these countries, which is unnoticeable.
My recommendation for this study at below;
- In the abstract and the text, TIMMS could not be understood. Please explain in the text and give the full name in the abstract.
- "GCC countries" is MesH terms? Is it correct? Please check.
- The introduction is well written. Please explain why you conduct this study among these countries and add the reason to your aim. Also, you need to explain this study's two main aims: evaluate the scale's psychometric features and compare the countries.
- How did you collect the data and reach the participants? Please explain this to the readers who do not know the TIMMS.
- The ethical consideration part should be added to the method section, and please provide the IRB date and numbers, the approval of the legal guardian, and adolescents.
- In the results section, is it possible to convert the table-1 figure? It must be clarified and understandable, which might help understand factor analysis from the numbers.
- The discussion part was clear, and the authors have written all potential and cultural reasons for the countries' differences.
- The references should be written according to journal rules.
I wish you success in your work.
Reviewer 3 Report
Thank you for the opportunity to review this paper. I liked the large sample size and the fact that the populations are sampled from an area of the world that is understudied. However, I was very confused about this paper for several reasons. The point of the paper, as I read it, was to examine the prevalence rates of cyberbullying perpetration across 6 middle eastern countries. That point was hidden amongst a large set of analyses that I am not sure were needed. I will elaborate my points below:
1. Why compare the prevalence rates across these 6 countries? Is there some theoretical reason why Qatar is supposed to be higher or lower on cyberbullying than Oman, for example? Whenever cross-cultural data is presented, there has to be a clear theoretical reason to expect (or not expect) differences across these countries. This was not detailed at all.
2. The authors claim that before they can figure out prevalence rates, they have to create their own measure and then check for invariance. While that is true, I had several questions: (a) why not use an already created measure and validate that, (b) there is no assessments of reliability or validity to show the measure is good, and (c) why not examine other predictors or outcomes of cyberbullying, as alluded to in 66-71.
3. The scale that is being tested is an 11 item scale that has 4 anchors. The scale range (if summed) is 11-44. How do the countries get a negative score (see Table 7). This is the biggest issue for me. I don't know what a score of -.877 means (for Qatar) on this scale. Further, I can't compare this score with scores from other research from other countries to get an understanding of where the middle east ranks with the US, Europe, Asia, etc.
4. Lines 100-101 say the TIMSS data group considered the measure unidimensional. Who is this group with such expertise? Further, why rely on their expertise when an Exploratory Factor Analysis can actually get at that answer. Examination of the items would lead me to disagree with this advisory group: some items focus on physical aggression (6, 8), others focus on relational aggression (1, 2, 3, 7), others focus on illegal behaviors (4, 5), and others are cyber (8, 9, 10), and one is verbal (11).
Overall, the need to understand the prevalence rates for Middle Eastern youth samples is massive; however, the way this study was done does not answer that question adequately nor does it provide any evidence that their questionnaire measures what it is supposed to.
Reviewer 4 Report
Congratulations to the authors for the work, it is very interesting and innovative.
I would start the abstract with some reference to the introduction and not directly to the objective.
In the methodology section, it would be convenient to indicate the percentage of both sexes, as well as the mean age and standard deviation for readers who do not identify the 4th grade at the international level.
It would be desirable to clearly state the conclusions in the article. It is referred to in the abstract but is not clear in the text.
Justification should be given as to what the theoretical process was for selecting the questions selected in the questionnaire.
It should be discussed to a greater extent, why multi-dimensional models are not used, e.g. bullying and cyberbullying.
Round 2
Reviewer 3 Report
This paper is much improved. With the goals laid out more clearly, the objectives of the study along with the analyses were more justified and make a better contribution.
Reviewer 4 Report
Congratulations to the authors for the improvements in the article. These have been carried out in depth and have given the article greater methodological rigour and a good theoretical underpinning. The results and the conclusion of the study have also been clarified.